Mixed effects of ecological intensification on natural pest control providers: a short-term study for biotic homogenization in winter wheat fields

Elek Zoltán zoltan.elek2@gmail.com 1
Růžičková Jana 1
Ádám Réka 2
Bereczki Krisztina 2
Boros Gergely 3
Kádár Ferenc 4
Kovács-Hostyánszki Anikó 2
Somay László 2 3
Szalkovszki Ottó 5
Báldi András 2 3
1 Biological Institute, MTA-ELTE-MTM Ecology Research Group, Pázmány Péter Sétány, Eötvös Loránd University , Budapest , Hungary
2 Institute of Ecology and Botany, Lendület Ecosystem Services Research Group, MTA Centre for Ecological Research , Vácrátót , Hungary
3 GINOP Sustainable Ecosystems Group, MTA Centre for Ecological Research , Tihany , Hungary
4 Plant Protection Institute, Centre for Agricultural Research, Hungarian Academy of Sciences , Budapest , Hungary
5 Department of Botany, National Biodiversity and Gene Conservation Center , Tápiószele , Hungary
Lin Brenda
Electronic publication date: 2020 Mar 27
Publication date: 2020
Volume: 8
Electronic Location ID: e8746
Received 2019 Sep 10; Accepted 2020 Feb 13
Copyright: ©2020 Elek et al.
Copyright year: 2020
Copyright holder: Elek et al.
License: This is an open access article distributed under the terms of the Creative Commons Attribution License, which permits unrestricted use, distribution, reproduction and adaptation in any medium and for any purpose provided that it is properly attributed. For attribution, the original author(s), title, publication source (PeerJ) and either DOI or URL of the article must be cited.
License URL: https://creativecommons.org/licenses/by/4.0/

Keywords: Aphids, Ecosystem services, Inorganic fertilizer, Pathogens, Set-aside field

Funding: EU FP7-project, LIBERATION (Linking farmland Biodiversity to Ecosystem services for effective ecological intensification 311781 MTA Lendület Program GINOP-2.3.2-15-2016-00019 This study was funded by the EU FP7-project, LIBERATION (Linking farmland Biodiversity to Ecosystem services for effective ecological intensification; project no. 311781) and the MTA Lendület Program. The analyses and writing of the paper was supported by the GINOP–2.3.2–15–2016–00019 project. The sampling design and the protocols were developed by the partners involved in EU FP7-project, LIBERATION (Linking farmland Biodiversity to Ecosystem services for effective ecological intensification; project no. 311781).

==============================
Agricultural intensification is one of the major drivers of biotic homogenization and has multiple levels ranging from within-field management intensity to landscape-scale simplification. The enhancement of invertebrate assemblages by establishing new, semi-natural habitats, such as set-aside fields can improve biological pest control in adjacent crops, and mitigate the adverse effect of biotic homogenization. In this study we aimed to examine the effects of ecological intensification in winter wheat fields in Hungary. We tested how pests and their natural enemies were affected at different spatial scales by landscape composition (proportion of semi-natural habitats in the surrounding matrix), configuration (presence of adjacent set-aside fields), and local field management practices, such as fertilizer (NPK) applications without applying insecticides. We demonstrated that at the local scale, decreased fertilizer usage had no direct effect either on pests or their natural enemies. Higher landscape complexity and adjacent semi-natural habitats seem to be the major drivers of decreasing aphid abundance, suggesting that these enhanced the predatory insect assemblages. Additionally, the high yield in plots with no adjacent set-aside fields suggests that intensive management can compensate for the lower yields on the extensive plots. Our results demonstrated that although complexity at the landscape scale was crucial for maintaining invertebrate assemblages, divergence in their response to pests and pathogens could also be explained by different dispersal abilities. Although the landscape attributes acted as dispersal filters in the organization of pest and pathogen assemblages in croplands, the presence of set-aside fields negatively influenced aphid abundance due to their between-field isolation effect.

Introduction

Landscape simplification has been suspected to influence the local patterns of species richness and abundance, because of the reduced capacity to support large species-pools, and the lack of opportunity for spill-over between different habitats (Tscharntke et al., 2012; Karp et al., 2018; Dainese et al., 2019). Nevertheless, the effects of agricultural intensification (characterized by intensive land use and the vast application of agrochemicals) at different spatial scales have only been partially analyzed (Hendrickx et al., 2007; Gámez-Virués et al., 2015; Gagic et al., 2017). Agricultural intensification may also serve as an ecological filter, simplifying entire communities through the process of biotic homogenization, as well as decreasing the diversity and resilience against the disturbance caused by farm management (Olden et al., 2004; Gámez-Virués et al., 2015; Tscharntke et al., 2016). The simplified arthropod communities might be restricted in their functions, and may result in the deterioration of ecosystem services (Yachi & Loreau, 1999; Woodcock et al., 2016). Although the environmental filtering of the local habitats and landscape composition may act as a driver for biotic homogenization, their effects on the assemblages of pests and their natural enemies has been rarely explored.

Insect pollinators, natural enemies, and soil decomposers are key factors for production in the various economically important crop systems (Bartomeus, Gagic & Bommarco, 2015; Klein et al., 2015; Tamburini et al., 2016). These groups provide regulatory ecosystem services (ES), often influenced by field management, landscape composition and configuration (Karp et al., 2018). There is an emerging interest in how agricultural management and external inputs can be combined with, or potentially replaced by ESs to enhance yields (Bommarco, Kleijn & Potts, 2013; Klein et al., 2015; Marini et al., 2015; Kleijn et al., 2019). Ecological intensification (EI) has been proposed with the aim to exploit the power of ESs in order to sustain agricultural production, while minimizing the adverse effects on the environment, such as the loss of biodiversity or landscape simplification (Kleijn et al., 2019). Although growing evidence suggests that ecologically intensified farming can safeguard food production and mitigate its adverse effects on the natural environment, the conscious use of ES providers (natural enemies, pollinators, decomposers, etc.) is still scarce (Bommarco, Kleijn & Potts, 2013; Martin et al., 2015). For instance, Tamburini et al. (2016) proved that biological pest regulation was influenced by soil management, suggesting that conservation tillage enhances soil fertility and natural pest control. Moreover, geographical bias exists: most of the studies on the relationship between farmland diversity (as a proxy for available ESs) and agricultural management have emerged from Western Europe (Tryjanowski et al., 2011; Sutcliffe et al., 2015) and North America (Kleijn et al., 2019).

In this study, we tested whether the enhancement of beneficial invertebrate assemblages, as pest control providers could contribute to the improvement of crop yields through reduced level of management intensity. We also looked for clues whether the number of newly established, semi-natural habitats adjacent to winter wheat fields could act as proxies for EI. We also tested whether fertilizers, as a key part of intensified crop management, interacted with landscape composition and configuration on selected pathogens, aphids and their natural enemies. We hypothesized that (i) a higher proportion of semi-natural habitats in the landscape can mitigate the negative effects of local (intensive) field management practices on the within-field abundance of natural enemies. Thus, enhanced connectivity between the habitat patches might lead to more natural enemies at the landscape level, and (ii) at the between-field scale, winter wheat fields with adjacent set-aside fields would have a higher abundance of natural enemies and enhanced natural pest control. At the plot level, we assumed that (iii) intensive inorganic fertilizer use in poor soils (acidic soil pH with low soil organic content) would enhance wheat yields, as well as the abundance of pathogens and aphids; while fertilizer usage in good soils (i.e., high soil organic content, neutral pH) would not have such an effect.

Material and Methods

Experimental area

Our study was conducted in the Heves Plain High Nature Value Area in North-Eastern Hungary. This region is one of the major target areas of agri-environmental schemes in Hungary (Kovács-Hostyánszki, Batáry & Báldi, 2011; Kovács-Hostyánszki & Báldi, 2012). The proportion of croplands in this region was 70%, the major crops being winter wheat, maize, as well as spring- and winter barley. Other major but less frequently sown crops were oilseed rape, sugar beet and sunflower. We selected 14 experimental winter wheat fields of 5–10 ha (geometric mean = 7.071 ha, S.D. = 3.53 ha). Seven of these had an adjacent, newly established (1 to 3 years old) set-aside field of sizes between 1.98 to 5.43 ha (geometric mean = 4.102 ha, S.D. = 1.65 ha), while the others, without any adjacent set-aside field, served as controls. The distance between the studied wheat fields was 1.54–10.84 km. The set-aside fields were sown by a seed-mixture, which included one leguminous (i.e., Medicago sativa) and two grass species (i.e., Lolium spp., Festuca spp.). During the maximum time of the 3-year set-aside management period, the use of agrochemicals was prohibited, and the fields were mown once a year in the second half of June (Kovács-Hostyánszki, Batáry & Báldi, 2011). The owners of these set-aside fields took part in the Hungarian Agri-Environmental Programme (HAEP-NHRDP 2007–13), in which the primary goal was to improve soil fertility and water retention, as well as to increase farmland biodiversity.

In 2014, we assigned an experimental plot of 45 × 20 m in each of the selected winter wheat field adjacent to the field margin (Fig. 1A). In the wheat fields with adjacent set-aside fields, the plot was at the edge towards the set-aside field. The farmers were asked to avoid applying fertilizers or insecticides within the experimental plots while maintaining all other conventional management practices. To assess the impact of fertilizer application and to avoid any interference between treatments, these areas were subdivided into two plots of 20 × 20 m, with a 5 m separation strip between them. One plot was treated with an NPK fertilizer at the usual rate of 95 kg N/ha in mid-April 2014, while the other plot received no fertilizer input (fertilizer control, Fig. 1A).

Figure 1 The experimental plots (grey squares) were established next to the field margin.

Within a winter wheat field (A), experimental plots (grey squares on panel) were established next to the field margin. The F + (fertilizer added) designate fertilized subplots, while the F − (no fertilizer added) is for the control ones. Within an experimental plot (B), the locations of inventory transects are indicated by white rectangles and subplots for wheat harvesting with black ones.

Soil characterization

We collected 15 cm soil cover to assess the soil organic content (SOC, a proxy for soil fertility) and pH. A total of 15 samples were taken per fertilizer control plots at the 10+ stage (BBCH scale, Zadoks, Chang & Konzak, 1974) of the wheat plants in mid-March and were stored at 4 °C. Prior to analysis, samples from each plot were sieved (at four mm mesh size) and blended. Subsequent soil analyses were conducted according to the Hungarian certifications (no. 08-0452 and 08-0206-2) and the method described by Mason (1983).

Landscape attributes

The landscape composition was calculated in a circle with 1,000 m-radius around each studied field. The proportion of arable fields, semi-natural habitats (semi-natural grasslands, semi-natural forests, tree lines, hedges and shrubs), urban areas and water bodies were measured using a GIS database (QGIS Development Team, 2018). We considered landscapes with >20% semi-natural habitats as structurally complex, while landscapes with <20% of such habitats were classified as simple (Batáry et al., 2011; Tscharntke, Batáry & Dormann, 2011). The presence/absence of set-aside fields adjacent to a study plot indirectly measures the spatial configuration (i.e., frequency of semi-natural habitats in the landscape) of the studied land use categories (Fahrig et al., 2011).

Arthropod and pathogen surveys

The abundance of aphids, their natural enemies and wheat pathogens was assessed by visual plant surveys in each plot along two parallel transects perpendicular to the margin, 5 m from each other at 5, 10 and 20 m from the edge of the wheat fields adjacent to the set-aside field, which were towards the habitat core. We randomly selected 17 wheat tillers per transect per distance (17 × 2 × 3 = 102 tillers per plot, Fig. 1B), and on each of these, we counted the number of grain aphids (mostly Sitobion avenae, Hemiptera: Aphidoidea), coccinellid adults and larvae (Coleoptera: Coccinellidae), syrphid larvae (Diptera: Syrphoidea), chrysopid larvae (Neuroptera: Chrysophidae), and the number of parasitized aphids (including mummies). We also visually assessed the major pathogens, including leaf spots (Septoria spp., Fusarium spp.), mildew and rust (Puccinia spp.) on the leaves. Plant surveys were conducted on three occasions in April (stem elongation stage; BBCH 35), May (head-forming stage; BBCH 50) and June (flowering stage; BBCH 60).

To monitor the activity density (a proxy for abundance from pitfall trap data, hereafter referred as ‘abundance’) of the surface-active natural enemies, four pitfall traps were installed along the transects (Fig. 1B) in each experimental (i.e,. fertilized and control) plot and operated for two weeks in late May–early June. The traps consisted of plastic cups of ∼114mm diameter, each of which filled with approximately 250 ml of a 50% propylene glycol solution saturated with NaCl and a drop of odorless detergent to reduce the surface tension. A green plastic roof protected the traps from litter and rain. The pitfall traps were emptied after 14 days, the catch was sorted in the laboratory, and the abundance of adult ground beetles and cursorial spiders (Lycosidae) was calculated.

Estimating yield

In June, all experimental plots were hand-harvested. Two sub-samples of 0.58 × 0.58 m area were taken at each distance (5, 10 and 20 m), resulting in a harvest sample of 1 m2 in total. Harvested ears were transported in cotton bags and dried at 24 °C for 48 h in a climate chamber under 10% RH. After threshing, we measured the total fresh grain mass per replicate batch, as well as fresh and dry mass of sub-samples in order to calculate the total grain yield (g/m2) (standardized to a moisture content of 14%); the grain and the ear mass were used for further analyses.

Statistical analyses

We used linear mixed models to test the responses of pests (aphid abundance), pathogens (prevalence of leaf spots expressed as percentage), natural enemies (abundance of ground beetles and spiders) and winter wheat yields (estimated as mass of wheat- ear and grain) to soil properties, nitrogen use, landscape complexity and configuration. To achieve a normal distribution of the residuals, all response variables were square-root transformed. We created sets of single-argument models on each response variable, to avoid collinearity and maximize the predictive power of explanatory variables, only one explanatory variable was considered in each model (Burnham & Anderson, 2002). As for explanatory variables, we included soil properties including pH (continuous variable) and SOC (continuous variable), nitrogen treatment (categorical variable with two levels: fertilizer applied/not applied), presence of adjacent set-aside field (categorical variable with two levels: with/without bordering set-aside field), or landscape complexity (categorical variable with two levels: simple/complex). In all models, the farm ID (nominal variable) was used as a random effect. For these aforementioned models we used the function ‘lme’ from the ‘nlme’ package (Pinheiro et al., 2017). Afterwards, each set of models was tested to select the best ones based on the Akaike Information Criterion (AICc) corrected for small samples sizes (Burnham & Anderson, 2002) using the ‘model.sel’ function from the ‘MuMIn’ package (Bartoń, 2016). When ΔAICc >2, the ‘best approximating’ model was selected as the most parsimonious explanation (see Table 1 for details). When more than one model had ΔAICc <2, we applied a model averaging approach for the uncertainty in model selection and obtained robust parameter estimates (Grueber et al., 2011). During model averaging, we built all possible models with the given explanatory variables and parameter estimates of the best models (ΔAICc <2), which were fully averaged with the models’ AICc weights. We used the ‘dredge’ and ‘model.avg’ functions from the ‘MuMIn’ package for model averaging. All the analyses were compiled in R 3.4.1 (R Core Team, 2019). The datasets collected on different sampling occasions were pooled for analyses. For all analyses, the alpha was set to 0.05.

Table 1 Summary of the model selection for each group.

Estimations are based on the calculated AICc value of the models, serving as the weight of evidence in favour of the different models. Most parsimonious models (delta < 2) are marked in bold.

Group	Models	df	logLik	AICc	Delta	Weight	
Pests and pathogens							
Aphids	set-aside	4	−825.814	1,659.7	0.00	0.735	
	pH	4	−827.027	1,662.2	2.43	0.218	
	SOC	4	−829.284	1,666.7	6.94	0.023	
	landscape	4	−829.868	1,667.8	8.11	0.013	
	nitrogen	4	−830.014	1,668.1	8.40	0.011	
Leaf spots	landscape	4	−573.825	1155.8	0.00	0.728	
	set-aside	4	−576.116	1,160.3	4.58	0.074	
	pH	4	−576.166	1,160.4	4.68	0.070	
	SOC	4	−576.217	1,160.5	4.78	0.067	
	nitrogen	4	−576.287	1,160.7	4.92	0.062	
Natural enemies							
Ground beetles	pH	4	−286.264	580.9	0.00	0.422	
	nitrogen	4	−286.910	582.2	1.29	0.221	
	set-aside	4	−287.467	583.3	2.41	0.127	
	landscape	4	−287.555	583.5	2.58	0.116	
	SOC	4	−287.578	583.6	2.63	0.113	
Spiders	landscape	4	−237.894	484.2	0.00	0.392	
	SOC	4	−238.766	485.9	1.74	0.164	
	set-aside	4	−238.799	486.0	1.81	0.159	
	nitrogen	4	−238.877	486.2	1.97	0.147	
	pH	4	−238.937	486.3	2.08	0.138	
Yield							
Ear mass	set-aside	4	−221.817	452.1	0.00	>0.999	
	SOC	4	−249.092	506.6	54.55	<0.001	
	pH	4	−250.217	508.9	56.80	<0.001	
	nitrogen	4	−253.255	514.9	62.88	<0.001	
	landscape	4	−253.528	515.5	63.42	<0.001	
Grain mass	set-aside	4	−208.626	425.7	0.00	>0.999	
	SOC	4	−236.602	481.6	55.95	<0.001	
	pH	4	−237.652	483.7	58.05	<0.001	
	nitrogen	4	−240.399	489.2	63.55	<0.001	
	landscape	4	−240.777	490.0	64.30	<0.001	

Results

Pests and pathogens

The presence of adjacent set-aside fields significantly affected aphid abundance. There were significantly more aphids in winter wheat fields without adjacent set-aside fields than in those with set-aside fields (Table 2, Fig. 2A). Neither soil properties, nor nitrogen use or landscape complexity affected aphid abundance. We did not find any significant effects of within- and between-field factors on the prevalence of aphids. The prevalence of leaf spots was higher in complex than in simple landscapes (Fig. 2B).

Table 2 Summary of the best models.

Coefficient and z values correspond to average model, while estimated (beta), t values and SD (random) are for the best models. Significant effects are in bold and directions of significant relationships (positive or negative) are designated by up and down arrows respectively. Marginal trends are underlined. Models explanatory power tested by R2 values and AICc.

Group	Best models	Variables	Coefficient/ estimated	SE	df	Z∕t	p	SD (random)	Marginal R2/fixed	Conditional R2/random	AICc	
Pests and pathogens												
Aphids	Set-aside	intercept	2.219	0.273	378	8.120	<0.001	0.872	0.113	0.276	1659.63	
		set-aside (with)	−1.4640.448	19	−3.2670.004	↓				
Leaf spots	Landscape	intercept	0.762	0.147	361	5.195	<0.001	0.493	0.059	0.232	1155.65	
		landscape (simple)	−0.5990.213	361	−2.8090.005	↓				
Natural enemies												
Ground beetles	Average	(intercept)	1.374	4.654		0.294	0.769		0.073	0.230	580.93	
		pH	1.211	0.665		1.706	0.088					
		nitrogen (yes)	−0.881	0.725		1.198	0.231					
Spiders	Average	(intercept)	3.872	0.573		6.673	<0.001		0.015	0.098	484.16	
		landscape (simple)	−0.800	0.551		1.360	0.174					
		SOC	−0.350	0.569		0.576	0.564					
		set-aside (with)	0.276	0.510		0.507	0.612					
Yield												
Ear mass	Set-aside	intercept	17.531	0.589	78	29.759	<0.001	2.140	0.597	0.820	451.61	
		set-aside (with)	−7.0340.715	78	−9.832<0.001	↓				
Grain mass	Set-aside	intercept	15.127	0.485	78	31.166	<0.001	1.716	0.619	0.810	425.25	
		set-aside (with)	−6.2020.615	78	−10.075<0.001	↓				

Figure 2 Response of aphid abundance (A), leaf spot prevalence (B) and winter wheat yield (C, D) to landscape configuration or complexity.

The yield was estimated by the ear (C) and grain (D) mass (g/m2). The portrayed values are means with whiskers representing 95% confidence intervals. Different capital letters above indicate significant differences.

Natural enemies

Our results did not indicate significant impact on the abundance of the two studied groups of ground-dwelling predators by any studied factors (Table 2, Fig. 3). However, the model selection procedure revealed that individual models, including pH and fertilizer treatments, were the two most parsimonious ones to describe the changes in carabid abundance, while those featuring landscape complexity and the presence of adjacent set-aside fields, as well as SOC and fertilizer treatments, were the best models for describing the changes in spider abundance (Table 1). In addition, soil pH had a marginally positive effect on the abundance of ground beetles, suggesting that, on average, slightly more beetles were present in the soil with neutral pH.

Figure 3 Response of the abundance for ground beetles (A, B), and spiders (C, D) to landscape configuration and complexity.

The portrayed values are means with whiskers representing 95% confidence intervals. Different capital letters above indicate significant differences.

Yield estimations

Ear and grain mass were significantly higher in fields without adjacent set-aside fields (Table 2, Figs. 2C, 2D) but no other variable had an impact on yield.

Discussion

We compared the effects of EI on the relationships between pests and their natural enemies at different spatial scales, from within- to between-field differences and up to landscape scale effects. We found that adjacent set-aside fields and landscape complexity seemed to be the drivers of the reduction in aphid abundance and the prevalence of pathogenic fungi. However, fertilizer use had no direct effect either on pests or their natural enemies.

Landscape influence on pests and pathogens

Agricultural landscapes are not static and may influence disease dynamics not only through its structure but also through its own dynamics (Plantegenest, Le May & Fabre, 2007). We found that the landscape configuration and complexity were key factors in the distribution of pests and pathogens, but the distribution of pathogenic fungi did not support the hypothesis that a higher proportion of semi-natural habitats in the landscape can mitigate the negative effects of local (intensive) field management practices on the within-field abundance of natural enemies, since complex landscapes did not restrict the distribution of pathogens. Although a previous study (Pfender et al., 2006) using a complex air pollution models (CALPUFF) demonstrated that the spores of pathogenic fungi had high deposition within a radius of 1–2 km around their source, the complexity of the landscape could actually lead to a locally higher infection rate due to a decreased air transfer distance of about 0.4 km (Plantegenest, Le May & Fabre, 2007). This finding was similar to our results on the prevalence of leaf spots, which was higher in the plots within the complex landscape. The decreased infection in simple agricultural landscapes could also be explained by the more effective chemical treatments (Plantegenest, Le May & Fabre, 2007; Gagic et al., 2017). We also demonstrated that the studied ground-dwelling predators seemed to be unaffected by the landscape constrains, leaving our hypothesis (i) unsupported. With regard to carabid beetles, in contrast to a previous study (Cole et al., 2002) we found that local species diversity in agricultural fields did not differ significantly when compared to semi-natural areas, owing to the high turnover of the typical agrobiont species. Our local spider assemblages were enhanced in the patches surrounded by a larger percentage of non-crop habitats, agreeing with earlier results by Clough et al. (2005) and Hendrickx et al. (2007). Additionally, the local communities in landscapes consisting of small and disconnected patches were characterized by a species composition with low beta-diversity. This homogenization caused by agricultural intensification may suggest that local assemblages are becoming more unsaturated, most probably because of the loss of specialist –and typically less competitive –species with low dispersal ability, such as many ground-dwelling predators (Clough et al., 2005; Hendrickx et al., 2007). In addition, highly mobile species could mask the effect of spatial heterogeneity between habitats, as demonstrated on orthopterans by Marini et al. (2011).

The beta diversity of natural enemies is masked by their mobility

We observed that less grain aphids were on plots that had adjacent set-aside habitats, while the natural enemies seemed unaffected, thus our hypothesis (ii) was just partially supported. This could be explained by the spillover of natural enemies into crop fields, resulting in better pest control (Woodcock et al., 2016). Any increase in aphid population growth, as well as any increase in aphid suppression with an increase in landscape complexity, could be explained by the higher availability of alternative resources and overwintering habitats in the semi-natural habitats around crop fields, benefiting both pests and their natural enemies (Martin et al., 2015; Karp et al., 2018). Additionally, the low abundance of aphids in the plots next to set-aside fields suggested that EI is also characterized by the contribution of other natural enemy groups to pest control (e.g., parasitoid wasps; Martin et al., 2015) or the isolation effect by the adjacent non-crop areas (Dainese et al., 2019; Karp et al., 2018).

We observed no effect of the presence of an adjacent set-aside field on the abundance of natural enemies. The available evidence (Hendrickx et al., 2007) suggests that the more specialized species (even natural enemies) abandon the isolated habitat patches that, consequently, contain only a few generalist species of high dispersal ability. The lack of response in two groups of natural enemies at the within-field scale might suggest that the intensive, asymmetric species flows (or spillovers) between the fields during the ripening phase of winter wheat masked eventual differences in habitat utilization between croplands and the semi-natural habitats (sensu lato Marini et al., 2011).

Yields: conditions within intensive fields are connected to local land-use traditions

Although we demonstrated no effect of fertilization on the abundance of pests, pathogens or their natural enemies, we found that the grain yield was higher in plots without adjacent set-aside than with them. In addition, the yield remained unaffected by soil pH or SOC. These results partially supported our hypothesis (iii), pointing to below- and above-ground ESs. At the below ground, the unaffected yield by soil parameters might be caused by the increased nitrogen loss rate in calcareous soils and the less efficient N utilization in the soil with intensive fertilizer application (Ju et al., 2009). A good quality topsoil with high SOC levels and optimal pH supports a healthy soil fauna (Scheu, 2001). A previous study by Pettersson, Tjallingii & Hardie (2007) revealed that higher abundance of grain aphids in fields with high SOC could be explained by their sensitivity to any changes in plant quality. Above ground, the observed lack of effect of fertilizer use on natural enemies, such as spiders and carabids, might imply that high SOC levels and optimal pH could support the soil fauna, which provides sufficient food for these groups to successfully resist management related disturbances. This may lie behind the marginally positive effect of pH on carabid abundance.

The fact that we did not detect any direct effect of the fertilizer on crop yield might lead to the conclusion that high soil fertility itself was more important than the external inputs. The lack of a relationship between the yield and any soil parameter such as pH and SOC could also be explained by the fact that while the best soils were farmed by intensive methods, the owners used their less fertile land as set-aside fields. We also found high yields in fields without adjacent set-aside fields. This could be explained by between-site variability in soil properties, which were independent of the proportion of arable land in the landscape, and being masked by intensive management practices (Williams & Hedlund, 2013; Bartomeus, Gagic & Bommarco, 2015; Gagic et al., 2017). Crop yield is influenced by a combination of biotic and abiotic factors. A better understanding of the interactions between above- and below-ground processes and ESs might ensure high crop yields, but this also requires the investigation of how they co-vary within and between crop fields (Gagic et al., 2017; Tamburini et al., 2016).

Conclusions

In conclusion, landscape features such as the presence of set-aside areas were the most important factors in determining the abundance of aphids in our studied cereal fields. We found that the lack of fertilization is not directly beneficial for spiders or ground beetles, suggesting that such extensive farming might not boost these natural enemies, due to the probability of increased between-field isolation caused by high compositional diversity at a landscape scale (Hendrickx et al., 2007; Martin et al., 2015; Tscharntke et al., 2016). Furthermore, our study focused on near- final ripening phase of the winter wheat, when the activity of natural enemies was the highest in the season, resulting in an equilibrium (equal abundance between croplands and adjacent set-aside fields) in habitat use between croplands, field margins and the set-aside fields (Tscharntke et al., 2012; Tscharntke et al., 2016). Therefore, conserving and enhancing the landscape diversity appears to be the key intervention to optimize the interaction between pests and their natural enemies in this agroecosystem. Our short-term study provided only a snapshot of the complex dynamics of agroecosystems; thus, the detected asymmetric species flows might have masked the differences in habitat utilization by natural enemies. Moreover, our demonstrative results underline the fact that agricultural practices can interfere with the abundance of natural enemies. Several landscape management practices may decrease the abundance of natural enemies and therefore the level of natural pest control. Hence, it is important to utilize evidence-based practices, which may include the reduction in chemical use, and promote natural enemies, as well as pest-specific measures to prevent the expansion of pest populations. Moreover, our results prompted us to conclude that management intensification at the within-field scale was still capable of compensating the yield gap caused by extensive management practices at the between-field scale.

Supplemental Information

Supplemental Information 1 Self-explanatory R codes for the data analyses

Click here for additional data file.

Supplemental Information 2 The interactive map of the sampling sites, Hever region, Hungary

The ESRI *.shp file can be opened in google earth or any GIS editing software. The attribute table includes the variables used in the analyses. The *.zip file should be extracted before use.

Click here for additional data file.

Supplemental Information 3 Stable 1: Aphid dataset for analyses

Referred to as ”(data = aphids)” in the R code file

Click here for additional data file.

Supplemental Information 4 Stable 2: Pathogens fungi (including leaf spots) dataset for analyses

Referred to as ”(data = spots)” in the R code

Click here for additional data file.

Supplemental Information 5 Stable 3: Carabids and spiders dataset for analyses

Referred to as “(data = pitfall)” in the R code

Click here for additional data file.

Supplemental Information 6 Stable 4: Wheat yiled dataset for analyses

Referred to as “(data = harvest)” in the R code

Click here for additional data file.

Supplemental Information 7 Description of the study sites in Heves region, Hungary

Click here for additional data file.

We thank all the farmers who allowed us to work on their land, László Tóth of the Bükk National Park Directorate for his help in the selection of the field sites as well as providing help during the study, Dorottya Molnár for field assistance, and Gábor Lövei for linguistic revision.

Additional Information and Declarations

Competing Interests

Author Contributions

Field Study Permissions

Data Availability

The authors declare there are no competing interests.

Zoltán Elek conceived and designed the experiments, performed the experiments, analyzed the data, prepared figures and/or tables, authored or reviewed drafts of the paper, and approved the final draft.

Jana Růžičková analyzed the data, prepared figures and/or tables, authored or reviewed drafts of the paper, and approved the final draft.

Réka Ádám, Krisztina Bereczki, Gergely Boros, Ferenc Kádár, László Somay and Ottó Szalkovszki performed the experiments, authored or reviewed drafts of the paper, and approved the final draft.

Anikó Kovács-Hostyánszki conceived and designed the experiments, performed the experiments, authored or reviewed drafts of the paper, and approved the final draft.

András Báldi conceived and designed the experiments, authored or reviewed drafts of the paper, and approved the final draft.

The following information was supplied relating to field study approvals (i.e., approving body and any reference numbers):

Field permits were not needed because field work was conducted on private land. Landowners (László Botos, Dániel Czakó, Attila Csala, László Mihály Csala, Zoltán Csőke, Vince János Gacsal, Antal Iván, Imre Kiss, Imre Koncz, Vince Koncz, Vince Lőrincz , Vince Nemes, Ervinné Németh, Ferenc Nyeste, Lajos Oláh, István Orbán, Jánosné Ördög, Miklós Sass, László Veres) provided permission to access their land.

The following information was supplied regarding data availability:

All data generated or analyzed during this study are available in the Supplementary Files.

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
