# Peer review of "Mixed effects of ecological intensification on natural pest control providers: a short-term study for biotic homogenization in winter wheat fields"

_PeerJ, doi:10.7717/peerj.8746_

## Round 0.1 · original submission · Major Revisions

Please take some time to find a science editor to help with style and organization of writing and language. All three reviewers have mentioned that some level of editing and reorganization will be necessary before publication. Additional clarity and reorganization is needed in the abstract and introduction, but also the title may be misleading. Be clear about what you are really measuring and studying.

Another important point made by the reviewers are improvements regarding the statistical analysis. Issues with sampling period as a random factor and collinearity should be considered. The reviewers provide a number of potential ways to help.

Please also take care to make sure that the tables and figures are stand alone and have all the required information necessary for a reader to understand them. There are some additional suggestions to make sure that the data within the supplementary materials are clear for others to be able to interpret as well.

Reviewer 1 ·

Basic reporting

-English language is, in general, good but needs some revision. Although I am not a native English speaker, I think that there is an unnecessary excess of “the” throughout the text (for example, lines 32, 33, 59, 166, 201, 236, among others). Furthermore, some complex sentences are difficult to follow or understand, such as in lines 69-70, 84-89, 103-106, 166-169, 299-303. Finally, more connection between some sentences in the introduction is needed. Therefore, I believe that a detailed revision of English grammar and an improvement of these pointed sentences can increase the quality of the paper.

-Introduction and background mention relevant references and all necessary topics are introduced. Nevertheless, I think that the first paragraph, where the authors describe the general overview of the background could be benefitted by a clearer organization from general (cropland expansion, agricultural intensification) to specific (landscape effects on animals, biotic homogenization) aspects. Besides, in the of this paragraph is not clear what the authors meant with “the current opinion on the effects of biotic homogenization…”. Are they trying to highlight the knowledge gap they studied? Probably no, because they mention community organization and this is not addressed here. Thus, I think that this last part could be changed to either emphasize the gap they are covering or at least improve the redaction.

On the other hand, EI is introduced in the third paragraph, but at the beginning of the second paragraph, they already mention that ecosystem services can replace external inputs and thus the concepts of EI are somehow mixed. I suggest that they focus the second paragraph on describing ecosystem services and their providers, and how landscape can influence them, and the third paragraph on EI.

-Structure of the article is good and follows the journal´s standards. Also, it can be considered as self-contained as well.

-Figures and tables are relevant and correctly labeled and in general, well described. Table 1 legend could be expanded to explain which information of the model selection process and each model is provided. Figures are, in addition, of good quality and clear. In figure 1 there is a small mistake in the legend, as it says that fertilizer is added in both F+ and F-. Also, it is not clear if the b panel is showing a zoom of one of the treatments, so perhaps it could be clarified in the text or showed in the figure with some arrows or other similar indication. In Figure 3, the drawings of the carabid and the spider could be added to panels a and c, respectively, to improve understanding. Also, in my opinion, the titles of Y-axis of figures 2 and 3 could be changed to “Aphid abundance (sqrt)”.

-Raw data and codes are provided with the submission as supplementary files and are clear. Nevertheless, I have problems to understand the codes of the farm names and how many data points per farm you had and in many cases, one code is mentioned at the beginning of the table and repeated later. I suggest changing this complex names for a simple number such as F1. In the case of data points, for many of the tables, the number of points per farm is not the same and it is not stated if some data were lost or it was not possible to obtain them. Finally, for aphid and pathogen data, a column stating the sampling date would be helpful too.

Experimental design

-The article reports the results from original research and fits well within the scope of PeerJ. Also, the investigation is performed following good technical and ethical standards.

-The main research question is considerably clear and the knowledge gap is established throughout the introduction. However, when the research question is described with more details, as in lines 103-109, I believe that the writing is not so clear. On one hand, the authors talk about “related pest control”, but pest control is not analyzed in the article. Therefore, they should explicitly say that they studied natural enemies or providers of pest control, but not the service per se. Specifically, in lines 103-104 they mention “enhancement of invertebrate assemblages” while they are meaning beneficial invertebrates. Concerning this, the title of the article mentions effects on natural pest control and I believe this should be changed too.

Secondly, the authors refer to the presence or absence of set-aside land as a measure of landscape configuration. In my opinion, this is not strictly true, because they addressed this presence as a local variable (set-aside neighboring the wheat field or not), whereas landscape configuration implies some measure at the landscape scale. Finally, while I think it is acceptable to talk about pests in general throughout the introduction, I think that in the objectives and hypotheses they should refer to aphids only, as this was the only group of pests studied and as it is written now it reads as they worked with many groups.

In connection with the research question, the hypotheses need some fine-tuning. In hypothesis 1, they mention an enhanced dispersal potential of natural enemies in relation to landscape composition. However, it may be better to refer to the connectivity in these landscapes that allows the dispersal, as it seems that they imply that dispersal abilities increase in complex landscapes. Or maybe they could talk about higher dispersal success. They also refer to higher pest control in this hypothesis, which was not directly evaluated. If they meant that this pest control could be reflected in pest abundance, they should say that clearly. Hypothesis 3 describes an interaction between fertilizer use and soil quality, but they did not test any interactions in their models, so they are not testing this hypothesis, which I believe it is very interesting. Please see below my concerns about statistical analyses for more details. Finally, no hypotheses for pests and pathogens are given, and no predictions for how yield can change at the landscape level (although they could expect an increase if pest control is higher, but this is not explicitly stated).

-Methods are described in detail and with sufficient information, though I have some minor comments. It would be good if they can include a figure showing a map of the study sites, at least as a supplementary figure. It is not clear if the with/without set-aside land treatment was assigned to seven pair of neighboring fields or if they are interspersed in the region. In connection to the field used for the study, they could also provide a supplementary table with the characteristics of each site in terms of proportion of semi-natural habitats. I consider that the separation of simple vs. complex landscapes is valid, but I have some doubts regarding the limit (less of more than 20%) because it can be the case that some of the simple landscapes are close to 20% and some of the complex too. Thus, these values are necessary for clarifying this issue and if there is not a clear separation between landscapes categories it may be a better idea to analyze landscape composition as a continuous variable. For the soil characterization, more details on the size of the soil cores and the distance separating these samples within each field.

Validity of the findings

-Data are of good quality and presented in their raw form, but I have some questions and suggestions about statistical analyses.
The authors stated that they used GLMMs but then transformed the data using square root. I believe that all these data could be used in the raw and analyzed with GLMMs with Poisson or negative binomial distributions using the same R package, except for harvest data which could be analyzed using a normal distribution without transformation if the data fit well. Although the square root may not be so bad as other transformations such as log, it still performs worst than using other distributions according to O´Hara and Kotze (2010).

My second concern with analyses is about data dependence. The authors used farm ID as a random factor, but this only considers dependence within each farm. For aphid and pathogens, they had several temporal repetitions that should be incorporated using sampling period as a random factor too. Furthermore, if there were multiple measurements within each fertilizer treatment, transect and/or distance to edge should also be considered, nested within farm.

Finally, I do not fully understand why the authors did not test models with more than one independent variable. I read that they wanted to avoid collinearity between variables, but many of the factors were under a crossed design and this should not be a problem, whereas soil characteristics variables could be included and Variance Inflation Factors calculated to test for collinearity. Moreover, one of the hypotheses was that soil fertility would interact with fertilizer use, and this should be tested using an interaction between these variables. I think that the model averaging approach they used is especially good for testing models with more than one variable and they could find some interesting results that are not covered by these single variable models. Also, a null model without any variables would be necessary to compare it against the other models.

-In general, the discussion is limited to the results and other related studies. The conclusions section could be shortened to focus specifically on the implications of their findings. I think that most of the conclusions section is based on their results except for the statement on lines 359-361 when they mentioned “detected asymmetric species flows” but they only worked with abundance, so I do not think this is the best way to describe it.

-Negative or non-significant results are well discussed and considered as important, so the authors did a good job avoiding bias towards significant differences.

Additional comments

In this interesting article, the authors investigated how two strategies of ecological intensification (EI) of agriculture (set-aside land and fertilizer use), landscape composition, and soil conditions affect a group of pests (aphids), pathogens, natural enemies, and yield of wheat crops. The topic is highly relevant and has implications for management applications of agricultural landscapes. However, I found several aspects of the manuscript that in my opinion should be improved before it is ready to be published. Below, I describe these issues in the five sections suggested by the journal. I apologize in advance for the extension of my comments, but I hope that this will help the authors for improving this interesting work.

-In addition to all my previous comments, I have one more general comment regarding the title of the study. Besides the natural pest control part that I already commented before, the title is focused on biotic homogenization and I do not think this is strictly investigated here. Usually, this topic is addressed by community composition or beta diversity between samples/sites. But I believe that in your study the main focus is not on biotic homogenization but pests/diseases, natural enemies and yield. I believe that studying the effects of EI on all these variables is valuable and you should highlight this in the title.

Other minor comments are provided below.
-Abundance of adult carabids was used as a proxy for natural enemies, but non-predator species of carabids can represent a high proportion of individuals in arable fields in Europe, such as species from the genus Amara that feed almost exclusively on seeds. Do you think that this could have biased your results?
-The specific objectives of your study should be clearly defined in the abstract. Instead of using terms as “such as” (line 33), you could mention directly the EI, landscape and soil factors tested and the analyzed groups.
-Line 43: dispersal ability might be a more appropriate word than power.
-Line 68: space is missing in et al.
-Line 79: the correct spelling for the reference in Bartomeus.
-Lines 98-101: you state that most of the evidence is from Western Europe but Sutcliffe et al. title is about central and Eastern Europe.
-Line 127: you could either provide the real average value for all fields or the range of sizes.
-Line 129: the distance was from … to … might be a better option.
-Lines 163: check the reference, publication year was 2011.
-Line 181: where were pitfall traps installed, specifically? Pitfall sampling was performed only once? At which moment?
-197: these sub-samples are not clear to me. Did you take sub-samples of known weights? What were the final measures used for analyses?
-Lines 232-233: Please refer to all the results about aphids first and then to pathogens.
-Line 232: The differences in Fig. 2b does not look so significant, CIs are quite large and overlapping. Is this result or graphic correct?
-Line 248: Moreover could be replaced by specifically or something similar because you are talking of the same results as in the previous sentence but with more details. Otherwise, you could remove the first sentence and directly mention the effects for both measures of yield.
-Line 249: It may be better to show yield results in a figure after the natural enemies Fig. 3, to follow the order of the results and tables. Alternatively, you could merge all pests and natural enemies into one figure and show yield in a second one.
-Lines 253-254: here you also mention the relationships between pests and enemies, which were not tested directly.
-Line2 259-261: if fertilizer use did not affect on wheat yield, I do not see how this within field intensification can compensate for the differences between fields. Please explain further.
-Lines 275-278: better than saying the EI can be described by, the authors could say that this effect of this particular EI measure could be linked to other enemies or dispersal barriers.
- Line 305: why you mention this potential scavenger role of predators and beta diversity in the title? I do not see the link with beta diversity in your study, whereas it is not clear if the mention of scavengers is linked to the seed detection you mention and how is this connected with your results.
-Line 364: “Better” is not an ideal adjective to describe natural enemies. Do you mean higher diversity, more individuals, or improved effectiveness?

References cited:
-O’hara, R. B., & Kotze, D. J. (2010). Do not log‐transform count data. Methods in ecology and Evolution, 1(2), 118-122.

Reviewer 2 ·

Basic reporting

There a few grammatical mistakes throughout the manuscript, although the text remains perfectly understandable. In the introduction, the connections between paragraphs and sentences are weak. Too often the authors resorted to conjunctive adverbs to add new details (I counted six “furthermore”, three “additionally/in addition” and one “moreover”). I think that too much information are added one after the other, and it is not clear what is most relevant for this work.

In the introduction ecosystem services are given by the acronym “ESs”, but the authors keep using the full name throughout the text. Also, US and UK spellings are mixed up (fertiliser vs. fertilizer, fertilisation vs. fertilization). Please, decide for one and be consistent.
* * *
Important references are missing from the introduction and from the discussion:

Dainese, Matteo, et al. "A global synthesis reveals biodiversity-mediated benefits for crop production." bioRxiv (2019): 554170.

Karp, Daniel S., et al. "Crop pests and predators exhibit inconsistent responses to surrounding landscape composition." Proceedings of the National Academy of Sciences 115.33 (2018): E7863-E7870.

Those are recent papers that are very relevant for this study and should have been at least discussed.
* * *
Tables are OK (maybe switching their order would be a good idea).
Figures could be improved. The captions are supposed to be self-explanatory but they are not always as such. For example, in Fig3 “Response of ground beetles (a, b), and spiders (c, d)” don’t clarify if it refers to abundance, density, species richness…
Please add the unit (“g”) in the vertical axis of Fig.2. The figures itself would be clearer if the mean is represented by a point rather than with a box.
* * *
The discussions would have been clearer if referring to the hypotheses (accepted, rejected, etc.) whenever relevant.

Experimental design

The manuscript would fit the aims and scope of the journal.
* * *
The research question is well defined, relevant and meaningful. However, the authors refer to natural pest control as if they measured it, and they are providing new insight about it (lines 69-70, 81-84, 90-95, 104, 111 in H1, but also lines 361-363 in the conclusions). Even the title mentions “the mixed effects ecological intensification on natural pest control...” and this is misleading. In this study, the authors measured the abundances of natural enemies (ground beetles and spiders) and of pests (aphids), which does not translate directly to measuring natural pest control. Tue authors may claim that the abundance of natural enemies could be considered as a proxy of natural pest control, but there is ample evidence that this is not always true. For example,

In barley fields: Rusch, Adrien, et al. "Predator body sizes and habitat preferences predict predation rates in an agroecosystem." Basic and Applied Ecology 16.3 (2015): 250-259.

In wheat fields: Mansion‐Vaquié, Agathe, et al. "Manipulating field margins to increase predation intensity in fields of winter wheat (Triticum aestivum)." Journal of Applied Entomology 141.8 (2017): 600-611.

Dainese, Matteo, et al. "A global synthesis reveals biodiversity-mediated benefits for crop production." bioRxiv (2019): 554170

Maybe the introduction should mention ecosystem services (ESs) without devoting excessive attention, as the only ES that was measured was crop yield. In fact, this experiment was not really designed to test H1 (on pest control), which is not followed up in the discussion.
* * *
The investigation was carried out accordingly to the prevailing ethical standards in the field.
* * *
Overall, the M&M section is OK. The authors should specify that leaf spot prevalence was calculated as the percentage of the plant with spots (was it?).

The statistical method used to analyse the data is good. The authors used multi-model inference and cited all the basic literature making clear what they are doing it. They also built models with only single explanatory variables to avoid potential collinearity between explanatory variables, which is a safe approach (although the reality may be more complicated than that). Nevertheless, it’s not clear if model averaging (in the two cases when it was applied) put together explanatory variables that were collinear. I am not aware of a solution to account for this, but the author should at least recognise it in the m&m or in the results. Did it happen that two collinear factors ended up together because of model averaging?

Validity of the findings

The parts were natural pest control is discussed should be removed, as natural pest control was not directly measured in this study.


Line 305. In the title of the subsection “The natural enemies are scavengers…”. It’s not clear what the authors mean, as they don’t discuss food preference and in any case spiders (that were also studied here) are not scavengers. The authors emphasise the spatial and temporal dynamics (spillover and seasonal effects) but neither of them was the focused of this study, which was only a “snapshot” of the season and that was not based on directional traps (which are particularly useful to determine spillover).

As I said before, the discussions would have been clearer if referring to the hypotheses whenever relevant. The authors spent more time discussing what they found to be the best models and the significant explanatory variables than discussing their hypotheses (for example, lines 257-259, the set-aside fields were expected to have an effect on natural enemies (H2), but the fact that no effect was found is not mentioned). I think the authors should speculate more about the reasons why their hypotheses were accepted/rejected. At least as much, if not more, than they speculated about what they found to be significant.

Additional comments

The introduction is confused. Starting by talking about biotic homogenisation immediately (lines 51-57 are unnecessary) would be an improvement. After discussing biotic homogenisation and how its effects on the community at different spatial scales have not been clarified (lines 64-65), the reader naturally expects to read what we know so far about the effects at within-field and landscape scales, but virtually all the attention is focused on the landscape scale.

Table 1 is already part of the results (the best models) and should be considered as such. I would suggest to switch Tables 1 and 2 and to start mentioning them from the results.
Another minor thing is that generalised linear mixed models should be changed to linear mixed models, as the authors normalised all responses and used always the Gaussian distribution.

Results could have been more informative, including mean and SD (or SE) when presenting the data.

There are several occurring mistakes in the reference list. The ones I detected are marked in the pdf.

Acknowledgments should not include the funders according to the guidelines of the journal.

Please find other, specific comments in the pdf of the manuscript.

Annotated reviews are not available for download in order to protect the identity of reviewers who chose to remain anonymous.

·

Basic reporting

Structurally I think this needs some work, although it has a lot of promoise. i would consider gettign a native english speaker to read though it before a resubmission. It needs a bit of polishing on this front. The abstract needs a bit of work though, see comments below.

Experimental design

See gerneal commetns to authro, however this si a farily sensibel split pot design. Smapling effor is not crazy (10 days of pitfall trapping - only 1 year of sampling) but there is reasonable field replication.

Validity of the findings

See gerneal commetns to author, in particualr the comments about the stats section., I feel you need to pick one statistical approach (probaly thre multimodel averaging). the including of this with the univariate correlations seems unecisairy. Note that if variabels are highly colinear jsut drop them enterly. Multimodel averaging is as suceptabel as any other approach to this issue - although I woudl point out thats its a rare ecologcial study where all covariates are totally indipendent - some correlation is common.

Additional comments

Abstract : In general this needs a lot of work for clarity. I would try and condense it more specifically into sections that define 1) what the problem is, 2) a simple summary of what you did methodologically; 3) what your key results were; and 4) the implications of this for sustainable intensification. Its kind of there but comes across as a bit muddled and as such is hard to read.
L26: Its not immediately clear what you mean by ‘alternative ecological intensification in croplands’ and how this relates to your opening line about varying levels of homogenisation – i.e. is your assumption that heterogeneity and sust. intensification are by definition going to occur simultaneously. I guess they could but not necessarily. Further I am not sure what the ‘alternative’ bit relates to – you explain ecological intensification fine.
L27-30: I think I would split this sentence in two.
L31: I think you need to be a bit more specific about what sust.int. management practices are being implemented.
L32’ ‘and local field management’
L33 ‘such as experimentally’
L33: what fertilisers, NPK, liquid manure, solid manure? Specify what you mean here.
L39-40: This needs to be clarified a bit - I am not entirely clear what you mean by this. From 36-41 the narrative suggests for fields with no fertiliser being close to set aside decreased aphid abundance -but for field s a distance from set aside the higher aphid densities required fertiliser to offset the reduction in yield. Is this correct? It neds work for clarity.
L43: Do you explicitly test this?

General question is reduced aphid numbers correlated with increased yields. In the UK at least the main issue with aphids is in terms of disease transfer, rather than direct effects on yield.

Introduction
L54: Why are crop lands hyper dynamic?
L54: given your paper focus also see papers linking landscape to predator - pest control attributes - Karp, D. S., et al (2018). Crop pests and predators exhibit inconsistent responses to surrounding landscape composition. Proceedings of the National Academy of Sciences, 115, E7863-E7870.
L59 ‘decreasing system biodiversity and……………against disturbance.’ Would also make clear what you mean by disturbance, is this say pesticide use or cultivation.
L65-69: See also Woodcock, et al. (2014). National patterns of functional diversity and redundancy in predatory ground beetles and bees associated with key UK arable crops. Journal of Applied Ecology, 51, 142-151.
L79: For actual demonstration of sust. Intensification see Pywell, et al (2015). Wildlife-Friendly Farming Increases Crop Yield: Evidence for Ecological Intensification. Proceedings of the Royal Society, Series B, 282, 20151740
L84-88: A very good point, as you point out in many situations in wheat this is the main factor reducing yield.
L92-95: I would re-write this sentence. Lose the bit about ‘been introduced by scientists’.
L92: Would not hurt to have some statistics in here about the value of natural pest control.
L102: Delete this sentence, and start the next one (L103) with ‘IN this study we tested whether….’
L105-106: This is quite muddled. Its hard to follow. I would split into two sentences (First sentence ends at L105 ‘intensity.’ – then go on to talk about semi-nat habitat)– but in general you need to work a bit on clarity. You sometimes have jumps of logic that are not necessarily wrong but for the reader they come out of nowhere.
L106: Given the fact this saves one word I would just use Ecological intensification rather than EI, its clearer.
L 108 ‘management) as it interacts with landscape…’
L108: I think the assessment of impacts on pathogens is a really important part of this work 9even if its no-sig). I would mention it more in the abstract.
L110: At some point earlier than this I would maybe define probably what you mean by ‘intensive’. I know what you mean, but I think its worth doing.
L111: IOk so this is one of those jumps in logic. What enhanced dispersal capacity of predators? Is this increased by semi-nat habitat or is this an innate property of them. If the latter this would seem to be only true in some cases as ‘nat enemies’ covers a massive range of taxa from highly mobile ballooning spiders to wingless ground beetles. You need to tighten up the text in relation to these. At the least justify what you mean.
L113: So your logic doesn’t really follow here from L111, effectively you are saying they are poor dispersers, and so you would only expect increased effects close to semi-nat habitat (a reasonable assertion).
L114: What type of fertiliser – inorganic, or manure, or liquid fertiliser etc. they potentially have very different effects.
Methods
L126: Might be useful to talk about common rotations, or length of rotations.
L131: Maybe give full mixture and sowing rate in a appendix
L139: But presumably in the control fields it was also at an edge (just there is no set aside for it to be close to).
L140: The fertiliser statement seem contradictory to that in L143 what are you say you split this area and add NPK.
L147: ‘We collected 15 cm soil cover to assess…’
L149: what’s the approx. date of this?
L151: what did you assess, e.g. organic carbon, phosphorous etc?
L159: ‘We considered those landscapes surrounding experimental fields with more than 20% semi-natural habitat as structurally complex’
L161-163: I am not clear what you are trying to say here. Also note that in the UK cover crops refer specifically to a situation when post harvesting a temporary mixture (eg. Oats or mustard) is sown to protect soils and increase fertility over the winter period before a spring sown crop is sown. Is this what you mean.
- A general thought, are all your wheats the first wheat in the rotation (normally the most productive) or a second one?
L175: delete ‘Moreover’
L185 –I thing you need to include sampling dates.
Stats:

L205: So why not use a poisson or negative binomial distribution rather than use a sqrt transformation
L205: We created sets of….
L209-212: You inver sampling suggests two sampling dates. From what you describe you summed the two sampling dates (e.g. for the aphids) so that you have a single value per sub-plot position along the transect.
L212: Ok so what random effects will depend on whether you included the points along the transect as individual data points, or whether you just summed these per sub-plot. If you have got individual transect points I think you need a sub-plot classified nested within farmID. If you have just summed across the transects (and across dates) then your current random effects look ok.
L216220: So I am happy for you to use the multimodel averaging approach. I think 2 AIc is a good cut off. However, from what I can tell (L206) you just have models with a single explanatory variable (e.g. landscape complexity). I think you should start with a maximal model with all your explanatory factors in it and use MuMin to assess all combination of these models (including the intercept) to derive you AIC delta 2 subset. If you have genuine concerns that two factors are collinear drop one and simply state that they were highly intercorrelated and so were not included in the same model. Ecological data always has some intercorrelations, but I appreciate there is a cut off on this. Looking at your explanatory effects there is nothing that jumps out to me as being crazy to include all in one model from which to get your subset for multi-model averaging. To be honest I think you may have done this but the L206 suggests this is not the case.
Results
L237: So if you are using multimodel averaging and a delta AIC subset this is not really compatible with the term significant as you are not undertaking a specific test of significance that produces an associated test statistic (e. F value) and p value. Reading this now I think you are undertaking a significance test on the single factor models described in L205. Firstly if you are doing this, what significance test are you undertaking. My more relevant point is why are you doing this. I would suggest that you pick one approach (probably the multi-model averaging approach) and just present those results. As I said if collinearity is a problem, this is a problem for your multimodel averaging approach as well so if there really are big issues here you need to drop explanatory variables.
- Gerneal point: So there are various test statistics worth presenting here to describe the importance of different explanatory variables in your detla2AIC subset. These focus on the summed wi score (although I think MuMin has a scaled version of this so that it adds to 1) and the importance score (which as I recall maybe the proportion of models in the delta 2 aic subset which include a given fixed effect. Worth having a look at paper Symonds, M. E., & Moussalli, A. (2011). A brief guide to model selection, multimodel inference and model averaging in behavioural ecology using Akaike’s information criterion. Behavioral Ecology and Sociobiology, 65(1), 13-21. Note as this approach is relative (e.g. you will always have best fit model, even if it’s a poor fit) it may be worth considering a R2 measure of fit for your global model. This paper also describes this.
Note earlier comments about not being clear if you are considering the transect data points separately, or different sampling dates (and how this affects you model). From what I can see from your results you have either just summed these are averaged them. In which case your random effects seem ok to me – but you still need to be clear what you have done.
General comments for Discussion.
So as some of the results are likely to be dependent on how you change your stat in response to my comments ill give some overarching comments here (also I have already given you lots of comments). I would work on adding more references in L253-267. I would possibly consider structuring the discussion in relation to your hypothesis (maybe, this is just an idea). I would potentially also consider if sampling effort or sample size may explain your absence of effects. Also see the points I make above about refocussing to include just your multimodel averaging results (I don’t see the point of both doing univariate p tests and this approach).

---

## Round 0.2 · Major Revisions

Although the manuscript has been improved substantially, Reviewer 1 and 2 have outlined where they would like more changes. Perhaps marking where you have made changes more clearly in the revision notes and ensuring that the reviewer does not have to refer to the other reviewers comments could help each reviewer see where their comments were addressed.

Reviewer 1 ·

Basic reporting

- I thank the authors for their modifications of the manuscript “The mixed effects of ecological intensification on natural pest control providers: a short-term study for biotic homogenisation in winter wheat fields”. The English language has improved considerably, although there are several sentences where a revision is necessary and UK and US English are still mixed in some parts. The objectives and hypotheses paragraph is one of the sections where the writing can be improved. The authors state that they revised the text with a language editor, but I think that an additional revision by a native English speaker will be necessary.

-Introduction and background
The first paragraph could be reduced considerably and that should not affect what the author is trying to express. I think that too much information is provided and some sentences are too complex. Furthermore, the authors end this paragraph stating that the “the environmental filtering effect of the local habitats or landscape composition may act as a driver for biotic homogenisation, their effects on the assemblages of natural pests and their enemies has not been explored, which are the most influential drivers for community organisation”. This statement is not true at all, both pests and natural enemies were studied in the context of local and landscape composition in agricultural landscapes and the authors cited some of the articles that did it. Maybe they are referring to something different, but it is not clear in its current form.
-The structure of the article is good and follows the journal´s standards. Also, it can be considered as self-contained as well.
-Figures and tables are improved, although check the comment below on the mix of multi-model and significance tests.
-Raw data and codes are provided with the submission as supplementary files and are clear. I understand that the authors want to keep the original variable names and codes as in their international project,

Experimental design

-The article reports the results from original research and fits well within the scope of PeerJ.
-The investigation is performed following good technical and ethical standards.
-The main research question is clear and the knowledge gap is highlighted. In the hypotheses sections, I still consider that it would be better to talk about aphids and not pests in general.
-In the hypotheses 3, they argue that they were referring to a logical interaction between soil quality and yield/fertilization and that a simple model can show how soil can influence yield. Nevertheless, I think that with their current analyses only one independent variable was included at each model, they cannot really detect what they are referring to. The effects of soil quality on yield can be identified, but if fertilization increases yield only in poor soils cannot be tested without analyzing the interaction between soil quality and fertilization. If the authors are not willing to test it, I think that the last part of the paragraph should be removed and they could refer to this in the discussion.
-Methods are described in detail and with sufficient information.
-The authors replied to the reviewers’ comments in detail and in some cases, I agree with their justification, although in others I still believe that changes should be made.
Particularly, two out of three reviewers suggested that it might be better to analyze the data starting from models with all the independent variables. I believe that this should be tried, as there are no strong reasons to suspect highly correlated factors and, as Rev.#2 mentioned, even averaging models with only one variable have a risk when this approach is used. Furthermore, the multi-model averaging approach and the significance tests are mixed (as pointed out in the comment BW.4.1), but their reply to this comment takes you to another comment that does not deal with this.
-I appreciate the inclusion of the map, but I still would like to see a supplementary table with the characteristics of each site in terms of proportion of semi-natural habitats.

Validity of the findings

The conclusions section should be shortened to show only what the authors found the its implications. As it is, to many references are there and the key message of the story is somehow lost.

Additional comments

-I noticed that some of the comments are replied with “Changed as suggested”, when the text does not show a substantial change in the direction of the comment. Furthermore, I understand that they numbered the reviewers’ comments for organization, but referring to different comments between reviewers is confusing and tedious. Moreover, is some of these cases the concern pointed out by one of the reviewers is not exactly the same, but the authors still state that this was already answered before. Thus, I believe that a new, careful revision of the text following all previous comments is necessary.

Other minor comments are provided below
-Lines 40-42: you state that extra fertilization can compensate for the low yields in plots with set-aside lands, but there were no differences between fertilizer treatments. Please revise, it is not clear
-Line 86: a parenthesis closes after field management but it was not opened before.
-Lines 226-227: what do the authors mean by saying that datasets collected twice were pooled into one database for analyses? Did they sum or averaged the values? Or they included both measures in the analyses? If so, a temporal random variable might be necessary, besides the field ID as a random variable.
-Lines 261-263: I repeat this comment because I do not see a difference with the previous version. If fertilizer use did not affect wheat yield, I do not see how this within field intensification can compensate for the differences between fields. Please explain further.
-Check the reference for Dainese et al. 2019, Sci. Adv. The authors’ list is longer than the one you show here.

Reviewer 2 ·

Basic reporting

The revised version is an improvement of the original submission; important references have been added, figures have been enhanced, and overall the text is easier to be read. However, there are still several issues that could further ameliorate the paper and should be considered, or at least addressed, by the authors. Few grammatical errors persisted (please, see comments in the pdf) and UK and US spell are still mixed (e.g. fertiliser vs. fertilizer).

I still believe that the introduction would be easier to read if starting directly with the concept of agricultural intensification (which should be defined at the first mention). The first three sentences (lines 52-59) only briefly state that agricultural landscapes are highly homogenous systems (or they often threatened by homogenisation), and that the animal community (indeed all biodiversity) relies on the environmental conditions at both local and landscape scales. This is all true but as it is is somehow trivial. Removing those three sentences would not compromise the introduction at all, and would actually make it lighter, clearer, and more concise.

Experimental design

The research question is well defined, relevant and meaningful.
* * *
The investigation was carried out accordingly to the prevailing ethical standards in the field.
* * *
Methods have been sufficiently described.

Validity of the findings

The discussion has been improved compared to the original version of the ms.

The conclusion should be more circumspect, as the authors did not measure biological control directly. A few changes in the text may enhance this section (please see attachment).

Additional comments

Figures were improved after accepting part of the original recommendations. It is not clear why in Table1 (summary of model selection for each group) nitrogen appears to affect significantly spiders, but this is not mentioned in Table2 (summary of the best models). I already pointed out this aspect in the round revision, so I assume that the authors have checked it and everything is correct. Yet, it seems to be an interesting result and maybe the authors could add one sentence about it in the discussion? (just a suggestion).
* * *
In the reference list there are still errors. For example, Coleoptera should be capitalised in
Kulkarni SS, Dosdall LM, Spence JR, Willenborg CJ. 2017. Seed detection and discrimination by
ground beetles (coleoptera: Carabidae) are associated with olfactory cues. PLoS ONE
12:e0170593. DOI: 10.1371/journal.pone.0170593.

and the first author name (Kovács-Hostyánszki) should be capitalised in

Kovács-hostyánszki A, Báldi A. 2012. Set-aside fields in agri-environment schemes can replace
the market-driven abolishment of fallows. Biological Conservation 152:196–203. DOI:
10.1016/j.biocon.2012.03.039.

Please check the comments on the pdf of the original submission, as those things were already indicated there.
* * *
According to the author guidelines, funders should not be mentioned in the acknowledge because there is a separate Funding Statement specifically meant for that. Please confirms this with the Editor.

Annotated reviews are not available for download in order to protect the identity of reviewers who chose to remain anonymous.

·

Basic reporting

So this is a second revision and I have just gone though your responses / rebuttles to the manuscript. Great stuff, I am more than happy with your responses both when you have accepted my suggestions and when you have justified ignoring them. I can see the other two reviewers have also been pretty thorough as well (I feel your pain having to go though all of these, inclugding mine - it happens to every paper I submit so I know the work this entails). Ultimately I think this has really improved the work and clarity. I am happy to recomend accept.

Ben Woodcock

Experimental design

see above

Validity of the findings

see above

Additional comments

see above

---

## Round 0.3 · accepted · Accept

Thank you for taking the time to have a language editor review and revise your manuscript. That is very helpful to ensure that there are no grammatical mistakes and that there is consistency in language. I think the introduction and discussion are much improved.